# T3SS translocon induces pyroptosis by direct interaction with NLRC4/ NAIP inflammasome

Yan Zhao[1,2], Hanshuo Zhu[1,3,4], Jinqian Li[2,5], Hang Xu[1,3,4], Li Sun[1,3,4]*

[1]CAS and Shandong Province Key Laboratory of Experimental Marine Biology, Institute of Oceanology; CAS Center for Ocean Mega-Science, Chinese Academy of Sciences, Qingdao, China; [2]Tsinghua University-Peking University Joint Center for Life Sciences, School of Basic Medical Sciences, Tsinghua University, Beijing, China; [3]Laboratory for Marine Biology and Biotechnology, Qingdao Marine Science and Technology Center, Qingdao, China; [4]College of Marine Sciences, University of Chinese Academy of Sciences, Qingdao, China; [5]NHC Key Laboratory of Tropical Disease Control, School of Tropical Medicine, Hainan Medical University, Haikou, China

*For correspondence:
lsun@qdio.ac.cn

Competing interest: The authors declare that no competing interests exist.

## eLife Assessment

This **important** study shows that Type 3 secretion translocons in Edwardsiella tarda and other bacteria activate the NAIP-NLRC4 inflammasome. The data from cellular and biochemical experiments showing that EseB is required for activation of the NLRC4 inflammasome are **convincing**. This paper is broadly relevant to those investigating host-pathogen interactions in diverse organisms.

**Abstract** Type III secretion system (T3SS) is a virulence apparatus existing in many bacterial pathogens. Structurally, T3SS consists of the base, needle, tip, and translocon. The NLRC4 inflammasome is the major receptor for T3SS needle and basal rod proteins. Whether other T3SS components are recognized by NLRC4 is unclear. In this study, using *Edwardsiella tarda* as a model intracellular pathogen, we examined T3SS–inflammasome interaction and its effect on cell death. *E. tarda* induced pyroptosis in a manner that required the bacterial translocon and the host inflammasome proteins of NLRC4, NLRP3, ASC, and caspase 1/4. The translocon protein EseB triggered NLRC4/NAIP-mediated pyroptosis by binding NAIP via its C-terminal region, particularly the terminal 6 residues (T6R). EseB homologs exist widely in T3SS-positive bacteria and share high identities in T6R. Like *E. tarda* EseB, all of the representatives of the EseB homologs exhibited T6R-dependent NLRC4 activation ability. Together these results revealed the function and molecular mechanism of EseB to induce host cell pyroptosis and suggested a highly conserved inflammasome-activation mechanism of T3SS translocon in bacterial pathogens.

## Introduction

The host innate immune system responds to pathogen-associated molecular patterns (PAMPs) and damage-associated molecular patterns (DAMPs) via multiple pattern recognition receptors (PRRs). Inflammasomes are a group of cytoplasmic PRRs that detect intracellular pathogens or disruptions in cellular homeostasis (*Christgen et al., 2020*). NLRP1, NLRP3, NLRC4, AIM2, and Pyrin are well-established PRRs that always combine with the adaptor protein ASC to form canonical inflammasomes,

which activate the effector protein caspase-1 (Casp1), leading to the processing and release of interleukin (IL) –1β and IL-18 (*Kanneganti, 2020*). Casp1 can also cleave and activate gasdermin (GSDM) D, which subsequently forms channels in the plasma membrane, eventually leading to a type of lytic programmed cell death called pyroptosis (*He et al., 2015*; *Shi et al., 2015*). In the non-canonical pathway, Casp4/5 (in humans) and Casp11 (in mice) are activated by bacterial lipopolysaccharide (LPS) and trigger GSDMD-mediated pyroptosis (*Shi et al., 2015*; *Kayagaki et al., 2015*). Of these inflammasomes, NLRC4 responds to Gram-negative bacterial ligands, primarily flagellin, and components of the T3SS apparatus, in a manner that requires an upstream immune sensor protein called NLR-family apoptosis inhibitory protein (NAIP), which interacts directly with the PAMPs (*Miao et al., 2010*; *Zhao et al., 2011*; *Reyes Ruiz et al., 2017*; *Kofoed and Vance, 2011*). Mice possess several NAIPs, each of which detects specific bacterial ligands, while humans possess only one functional NAIP (hNAIP) that is capable of broadly recognizing multiple bacterial ligands (*Zhao et al., 2011*; *Reyes Ruiz et al., 2017*; *Kofoed and Vance, 2011*; *Yang et al., 2013*).

*Edwardsiella tarda* belongs to the Enterobacteriaceae family. It is an intracellular pathogen and can survive and replicate in host immune cells, such as macrophages (*Sui et al., 2017*; *Li et al., 2019*). *E. tarda* has a broad range of hosts, including fish and humans (*Leung et al., 2012*; *Leung et al., 2019*). In humans, *E. tarda* has been reported to cause gastrointestinal diseases and systemic infections that can be lethal (*Hirai et al., 2015*; *Leung et al., 2022*). *E. tarda* possesses T3SS and uses it to modulate the host immune systems (*Leung et al., 2012*; *Leung et al., 2022*). T3SS functions as an injectisome that delivers bacterial effector proteins into host cells. The T3SS apparatus consists of three distinct parts— the extracellular segment, the basal body, and the cytoplasmic components (*Portaliou et al., 2016*). The extracellular part comprises the needle, tip, and translocon, which spans the host cell membrane (*Dey et al., 2019*). In *E. tarda*, the translocon complex is formed by EseB, EseC, and EseD (*Tan et al., 2005*). Genetically, the *eseB*, *escA*, *eseC*, and *eseD* genes clustered in tandem in the same operon. In function, EscA acts as a specific chaperone for EseC (*Wang et al., 2009*). The translocon is known to be essential for the pathogenesis of *E. tarda* (*Tan et al., 2005*; *Wang et al., 2009*; *Okuda et al., 2009*), but the mechanism, in particular, that is associated with inflammasome activation and pyroptosis, remains to be explored.

In this study, using *E. tarda* as an intracellular pathogen model and human macrophages as the host cells, we investigated the function and the working mechanism of the T3SS translocon in pathogen-host interaction. We found that *E. tarda* induced GSDMD-dependent pyroptosis involving both canonical and non-canonical inflammasomes, and that the translocon proteins were indispensable for *E. tarda* cytotoxicity. We examined the role and mechanism of EseB in host interaction and uncovered the key structure of EseB that was essential for binding and activating the NLRC4/NAIP inflammasome. Furthermore, we identified EseB homologs in a broad range of bacteria and demonstrated that NLRC4/NAIP interaction and activation was probably a conserved function of the EseB homologs in T3SS-positive bacterial pathogens. These results added new insight into the working mechanism of EseB and highlighted the important role of the translocon in bacteria-host interaction.

## Results

### *E. tarda* triggers GSDMD-dependent pyroptosis in human macrophages

To examine whether *E. tarda* infection triggered cell death in human macrophages, differentiated THP-1 cells (dTHP-1 cells) were infected with *E. tarda*. The cells were found to undergo rapid cell death as indicated by LDH release (*Figure 1—figure supplement 1A*). However, cell death was almost completely blocked when the cells were pre-treated with cytochalasin B (CytoB) or cytochalasin D (CytoD), which inhibited bacterial entry into the cells (*Figure 1—figure supplement 1*). Hence, it was intracellular *E. tarda*, rather than extracellular bacteria, that induced cell death in human macrophages. Further examination showed that *E. tarda*-infected dTHP-1 cells exhibited a swelling morphology, accompanied by IL-1β release, Casp1 activation, and GSDMD cleavage (*Figure 1A–E*). These observations indicated that *E. tarda* triggered pyroptosis in dTHP-1 cells. To examine whether and which inflammasomes were involved in this process, cells defective in various inflammasome pathways were employed. The results showed that following *E. tarda* infection for 2 hr or 4 hr, cell death and IL-1β release were significantly decreased in NLRC4 knockout (NLRC4-KO) cells, Casp4 knockout (Casp4-KO) cells, and NLRP3 knockdown (NLRP3-KD) cells (at 4 hr post-infection), but not in Aim2

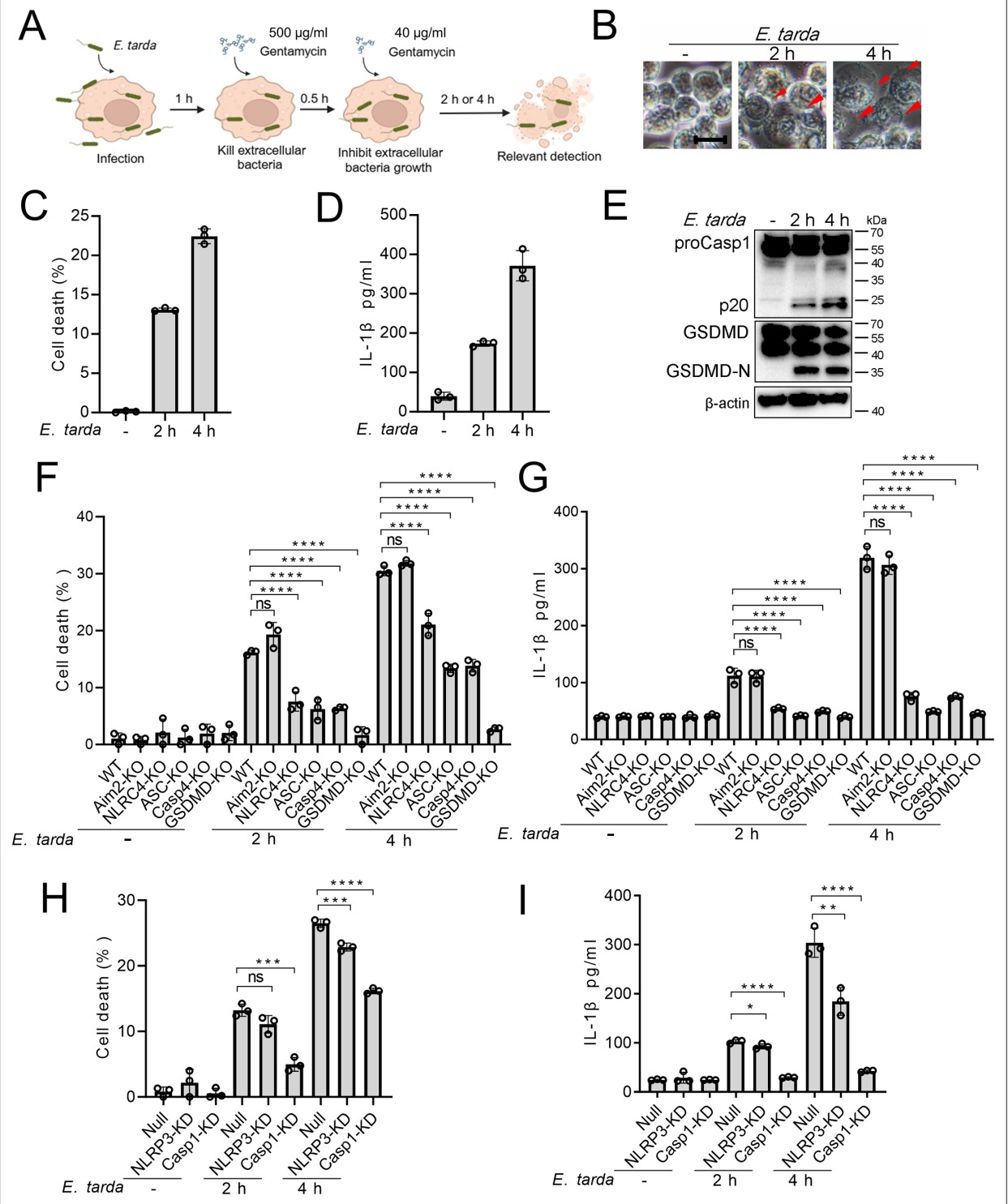

**Figure 1.** The ability of *Edwardsiella tarda* to induce pyroptosis in human macrophages. (**A**) The schematic of experimental design. (**B–E**) Differentiated THP-1 (dTHP-1) cells were infected with *E. tarda* for the indicated hours and then subjected to microscopy (**B**), measurement of cell death (**C**), IL-1β release (**D**), and Western blot (**E**) using antibodies against Casp1, GSDMD, and β-actin (loading control). In (**B**), red arrowheads indicate pyroptotic cells; scale bar, 10 µm. (**F–I**) dTHP-cells in the form of wild-type (WT), knockout (KO) variants (Aim2-KO, NLRC4-KO, ASC-KO, Casp4-KO, and GSDMD-KO), and knockdown (KD) variants (NLRP3-KD and Casp1-KD) were infected with or without *E. tarda* for 2 or 4 hr, and then assessed for cell death (**F, H**) and IL-1β release (**G, I**). For panels C, D, and F-I, data were the means of triplicate assays and shown as means ± SD. ns, not significant, ***p<0.001, ****p<0.0001, one-way ANOVA with Dunnett's multiple-comparison test.

*Figure 1 continued on next page*

*Figure 1 continued*

The online version of this article includes the following source data and figure supplement(s) for figure 1:

**Source data 1.** PDF file containing the original blots for *Figure 1E*.

**Source data 2.** Original files for blots are displayed in *Figure 1E*.

**Source data 3.** The numerical source data corresponds to *Figure 1*.

**Figure supplement 1.** The effect of cytochalasin B (CytoB) and cytochalasin D (CytoD) on the ability of *Edwardsiella tarda* to induce cell death.

**Figure supplement 1—source data 1.** The numerical source data corresponds to *Figure 1—figure supplement 1*.

knockout (Aim2-KO) cells (*Figure 1F–I*). Cells with Casp1 knockdown (Casp1-KD), GSDMD knockout (GSDMD-KO), and ASC knockout (ASC -KO) all exhibited significantly decreased cell death and IL-1β release (*Figure 1F–I*). Taken together, these results indicated that intracellular *E. tarda* induced GSDMD-dependent pyroptosis in human macrophages in a manner that required NLRC4, NLRP3, ASC, Casp1, and Casp4.

## The T3SS translocon is essential to *E. tarda*-induced pyroptosis

Similar to most intracellular Gram-negative bacterial pathogens, *E. tarda* possesses a T3SS system and uses it as a weapon against host immunity. In this system, EseB, EseC, and EseD form a translocon, with EscA acting as an EseC chaperone. To investigate the potential role of the translocon in pyroptosis, a series of *E. tarda* mutants were constructed, which bear the knockout of *eseB*, *escA*, *eseC*, or *eseD* (ΔeseB, ΔescA, ΔeseC, or ΔeseD, respectively), or the knockout of all of the four genes (ΔeseB-D) (*Figure 2A*). Compared with the wild-type, these mutants showed no deficiency in host cell adhesion or entry (*Figure 2B and C*). However, ΔeseB-D, ΔeseB, ΔeseC, and ΔeseD were unable to induce host cell death, IL-1β release, Casp1 activation, or GSDMD cleavage following infection (*Figure 2D–F*). Casp4 activation was also absent in ΔeseB-D-infected cells (*Figure 2—figure supplement 1*). ΔescA was still able to induce pyroptosis, but the levels of cell death and IL-1β release were significantly lower than that induced by the wild-type (*Figure 2D–F*). To examine whether flagellin was required for *E. tarda*-induced pyroptosis, the flagellin mutant, ΔfliC, was created. Compared with the wild-type (WT), ΔfliC exhibited no significant change in host cell adhesion/entry, cell death induction, Casp1 activation, or GSDMD cleavage (*Figure 2—figure supplement 2*). These results indicated that the T3SS translocon proteins, rather than flagellin, were indispensable for *E. tarda*-induced pyroptosis.

## Cytosolic EseB triggers pyroptosis in a NLRC4/NAIP-dependent manner

To examine the mechanism underlying the above observed essentialness of the translocon in *E. tarda*-induced cell death, the recombinant proteins of EscA, EseB, EseC, and EseD were prepared (*Figure 3—figure supplement 1*). The effects of these proteins, both extracellular and intracellular, on THP-1 cells were determined. When present in the extracellular milieu, none of these proteins caused an apparent change to the cell morphology (*Figure 3A*). However, when EseB was present in the cytoplasm of THP-1 cells, pyroptotic cell death, including activation of Casp1 and GSDMD, was observed (*Figure 3A–C*). To examine which inflammasome pathway was involved in this process, the effect of EseB was determined using Aim2/NLRC4/ASC/Casp4/GSDMD knockout cells and NLRP3/Casp1 knockdown cells. The results showed that defective in NLRC4, GSDMD, and Casp1, but not in AIM2, Casp4, or NLRP3, rendered the cells markedly immune to the death-inducing effect of EseB (*Figure 3D and E*). ASC knockout also significantly, though to a relatively modest extent, reduced EseB-induced cell death. Since NAIP is known to be involved in NLRC4 inflammasome activation, the effect of NAIP knockdown on EseB cytotoxicity was examined. The result showed that the NAIP-knockdown cells exhibited significantly decreased death following EseB treatment (*Figure 3F and G*). Collectively, these results indicated that cytosolic EseB, rather than extracellular EseB, induced pyroptosis via the NLRC4/NAIP inflammasome.

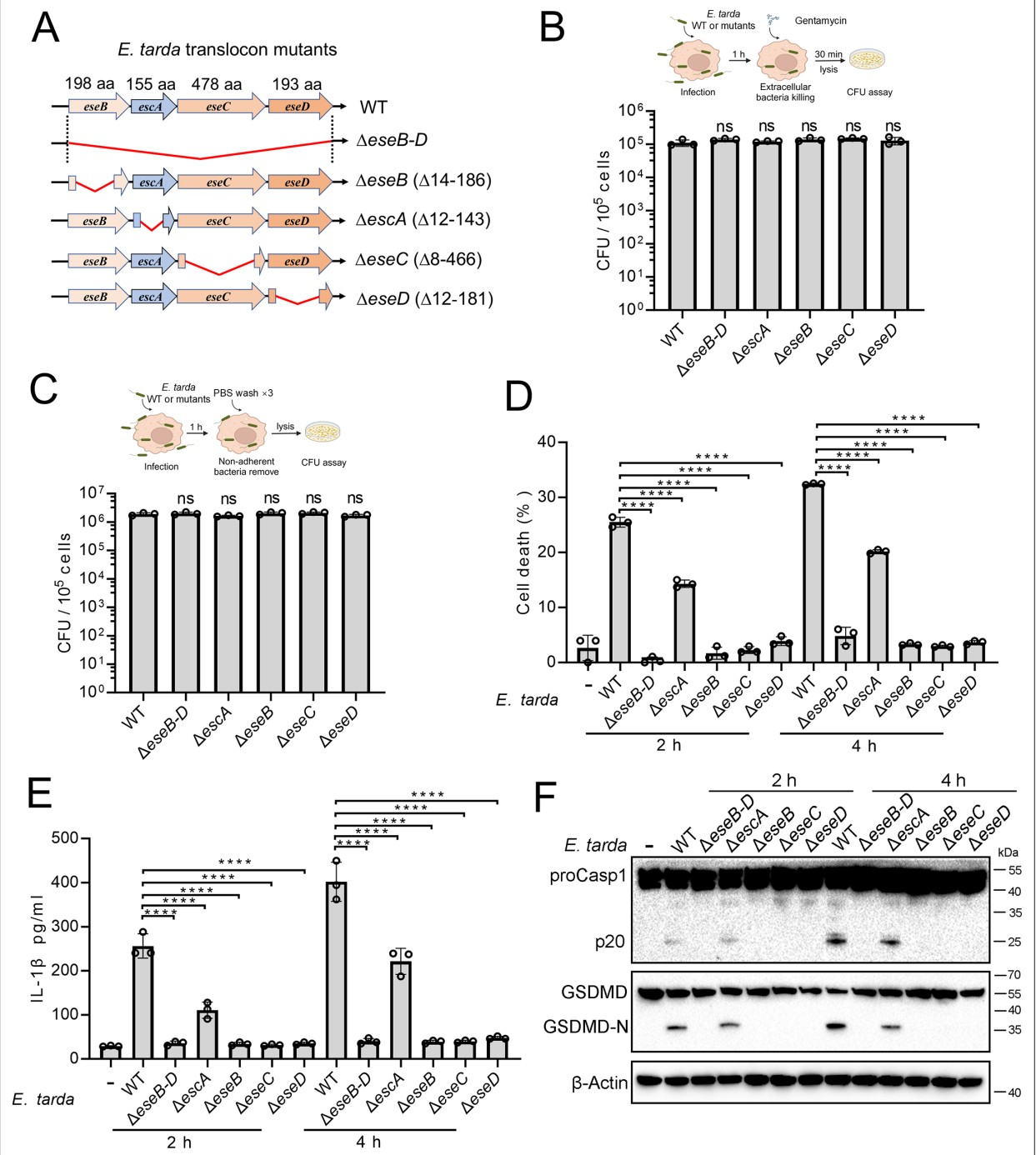

**Figure 2.** The importance of the translocon for *Edwardsiella tarda*-induced pyroptosis. (**A**) A diagram showing the in-frame deletion (red curved line) of *eseB-D, escA, eseB, eseC,* and *eseD*. (**B, C**) Differentiated THP-1 (dTHP-1) cells were infected with wild-type (WT) or mutant *E. tarda* for 1 hr. The intracellular bacteria (**B**) and the total bacteria associated with the cells (i.e. both the cell-attached and the intracellular bacteria) (**C**) were determined by plate count. (**D–F**) dTHP-1 cells were treated with or without *E. tarda* variants for 2 or 4 hr, and then subjected to cell death analysis (**D**), IL-1β release measurement (**E**), and immunoblot (**F**) using antibodies against Casp1, GSDMD, and β-actin (loading control). For panels B-E, data are the means of triplicate assays and are shown as means ± SD. ns, not significant, ****p<0.0001, one-way ANOVA with Dunnett's multiple-comparison test.

The online version of this article includes the following source data and figure supplement(s) for figure 2:

**Source data 1.** PDF file containing the original blots for *Figure 2F*.

**Source data 2.** Original files for blots are displayed in *Figure 2F*.

**Source data 3.** The numerical source data corresponds to *Figure 2*.

*Figure 2 continued on next page*

*Figure 2 continued*

**Figure supplement 1.** The effect of Δ*eseB-D* on Casp4 activation.

**Figure supplement 1—source data 1.** PDF file containing the original blots for *Figure 2—figure supplement 1*.

**Figure supplement 1—source data 2.** Original files for blots are displayed in *Figure 2—figure supplement 1*.

**Figure supplement 1—source data 3.** The numerical source data corresponds to *Figure 2—figure supplement 1*.

**Figure supplement 2.** The involvement of flagellin in *Edwardsiella tarda*-induced pyroptosis of human macrophages.

**Figure supplement 2—source data 1.** PDF file containing the original blots for *Figure 2—figure supplement 2*.

**Figure supplement 2—source data 2.** Original files for blots are displayed in *Figure 2—figure supplement 2*.

**Figure supplement 2—source data 3.** The numerical source data corresponds to *Figure 2—figure supplement 2*.

## The C-terminal region of EseB is vital to NAIP interaction and pyroptosis induction

To explore its mechanism to activate the NLRC4/NAIP pathway, EseB was first subjected to sequence analysis. The C-terminal (CT) region of EseB exhibits notable degrees of conservation, especially in the terminal 6 residues (T6R), with bacterial needle proteins known to activate NLRC4 (*Figure 4A*). Based on this observation, we divided EseB into the N-terminal (NT) (1–112 aa) and the CT (113–198 aa) regions (*Figure 4B*). To identify the sequence essential to EseB function, a series of EseB mutants were constructed that bear deletion of the terminal 4 residues (T4R) (EseBΔT4R) or the T6R (EseBΔT6R), or contain only the NT (EseB-NT) or CT (EseB-CT) region (*Figure 4B*, *Figure 4—figure supplement 1*). When introduced into THP-1 cells, EseB-CT induced cell death and Casp1/GSDMD activation, whereas all other EseB mutants failed to do so (*Figure 4C–E*). The EseB variants were further examined in 293T cells with reconstituted NAIP/NLRC4 inflammasome, in which NAIP/NLRC4 activation could be monitored by analyzing the maturation cleavage of IL-1β (*Figure 4F*). The results showed that in cells co-expressing EseB and all NLRC4 inflammasome components, massive IL-1β cleavage occurred (*Figure 4G*). Similar observations were made with cells co-expressing NAIP/NLRC4 inflammasome and EseB-CT (*Figure 4—figure supplement 2A*). Together these results demonstrated that EseB, via its CT region, activated the NAIP/NLRC4 inflammasome. Consistently, immunoprecipitation revealed that EseB, as well as EseB-CT, bound to NAIP, and this binding was not observed with either of the other EseB mutants (*Figure 4H, I*). EseB could not bind to NLRC4 directly (*Figure 4H*). In addition to EseB, the rod (EsaI), needle (EsaG), and flagellin (FliC1/2) of *E. tarda* were also examined for their ability to activate the NLRC4 inflammasome. For this purpose, these proteins were each co-expressed with NAIP/NLRC4 in 293T cells. Only the needle protein EsaG induced significant IL-1β cleavage (*Figure 4—figure supplement 2B, C*). This result indicated that EsaG, but not the rod protein or flagellin, activated NLRC4.

## The NLRC4-activation capacity and mechanism of EseB are conserved in pathogenic bacteria

With the above results, we wondered whether the observed EseB function was unique to *E. tarda* or common in bacterial pathogens with T3SS. To answer this question, we searched and identified EseB homologs in diverse pathogenic bacteria. Among these EseB homologues, 20 were randomly selected, including those from *Salmonella* and *Chromobacterium*, for activity analysis (*Supplementary file 2*). The results showed that all of the 20 EseB homologs could activate the NLRC4/NAIP inflammasome in reconstituted 293T cells (*Figure 5A*). Phylogenetic analysis of these EseB homologues and *E. tarda* EseB showed that they fell into four groups (*Figure 5—figure supplement 1*). However, high levels of sequence identities are shared among these EseB at the T6R (*Figure 5B*). To examine whether, as observed with *E. tarda* EseB, the T6R was functionally important, five of the 20 EseB homologs were selected for mutation to remove the T6R or T7R. The resulting mutants, like *E. tarda* EseBΔT6R, no longer possessed the ability to activate the NLRC4 inflammasome and cause cell death (*Figure 5C, D*, *Figure 5—figure supplement 2*). We further examined the NLRC4-activating potential of bacterial translocators with relatively low sequence identities (<23%) with EseB. For this purpose, 14 key translocator proteins from eight pathogenic bacteria were selected for the examination (*Supplementary file 2*). Three of these proteins, i.e., PcrV, SipC, and IpaC, possess terminal 5–7 residues that are similar to the T6R of EseB (*Figure 5—figure supplement 3A*). Subsequent study

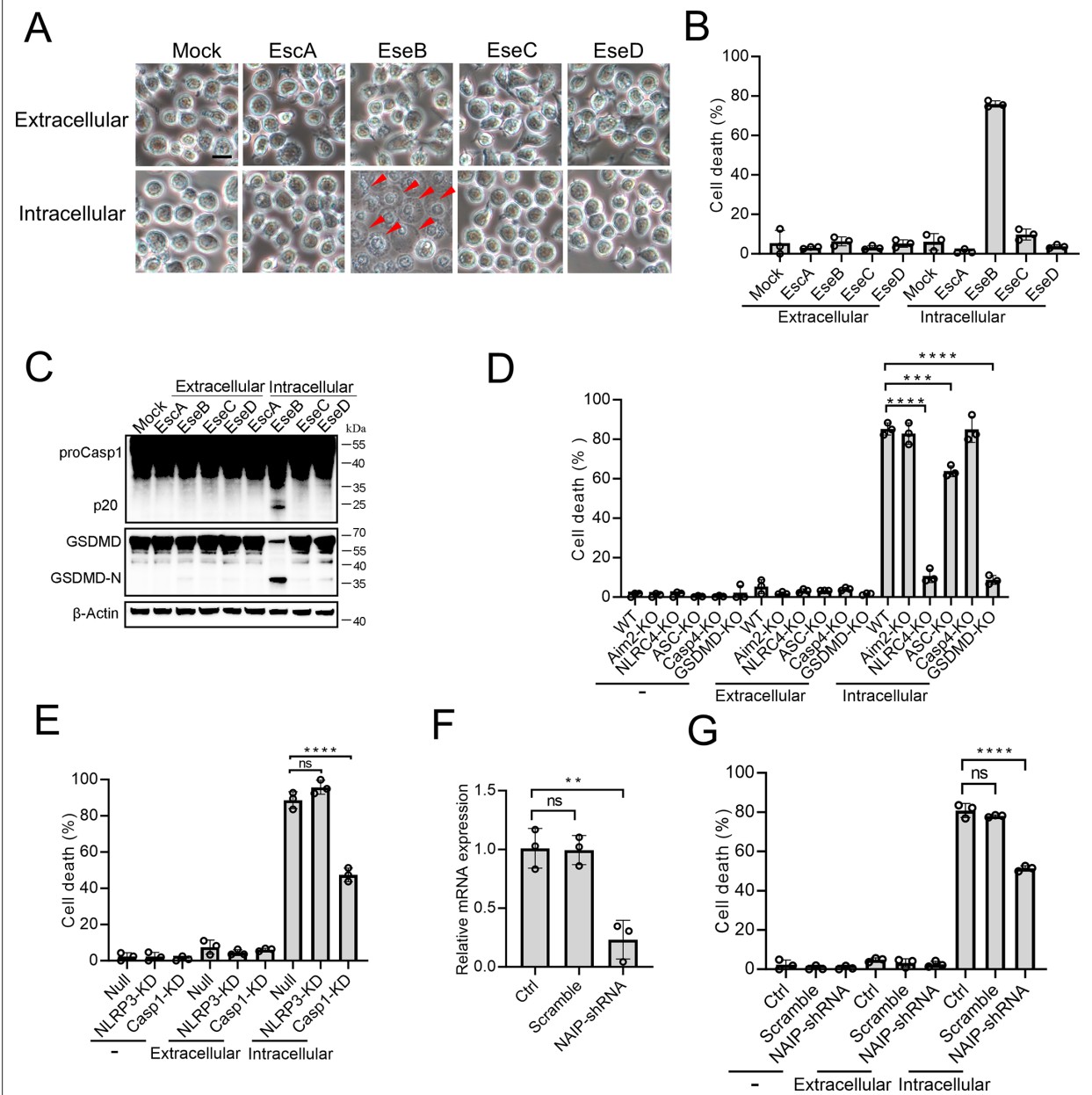

**Figure 3.** The pyroptotic effect of the translocon proteins and their dependence on the inflammasomes. (**A–C**) To determine the extracellular and intracellular effects of EscA, EseB, EseC, and EseD, each of the proteins was added into the culture medium of THP-1 cells (extracellular) or electroporated into THP-1 cells (intracellular). The control cells were mock-treated with PBS. The cells were subjected to microscopy (**A**), cell death analysis (**B**), and immunoblot using antibodies against Casp1, GSDMD, and β-actin (loading control) (**C**). In (**A**), red arrowheads indicate pyroptotic cells; scale bar, 10 μm. (**D**) The wild-type (WT) and knockout (KO) THP-1 cells were treated with or without extracellular and intracellular EseB as above and then examined for cell death. (**E**) The control THP-cells (Null) and the NLRP3/Casp1 knockdown (KD) THP-cells were treated with or without extracellular and intracellular EseB as above and then examined for cells death. (**F**) THP-1 cell treated with or without (Control, Ctrl) NLR-family apoptosis inhibitory protein (NAIP)-targeting shRNA or scramble RNA (negative control RNA) were examined for NAIP expression by qRT-PCR. (**G**) THP-1 cells administered with or without NAIP-targeting or scramble RNA were treated or without extracellular and intracellular EseB as above and then examined for cell death. For panels B, and D-F, data are the means of triplicate assays and are shown as means ± SD. ns, not significant, ***p<0.001, ****p<0.0001, one-way ANOVA with Dunnett's multiple-comparison test.

The online version of this article includes the following source data and figure supplement(s) for figure 3:

**Source data 1.** PDF file containing the original blots for *Figure 3C*.

**Source data 2.** Original files for blots are displayed in *Figure 3C*.

**Source data 3.** The numerical source data corresponds to *Figure 3*.

*Figure 3 continued on next page*

*Figure 3 continued*

**Figure supplement 1.** SDS-PAGE analysis of purified recombinant proteins.

**Figure supplement 1—source data 1.** PDF file containing the original gels for *Figure 3—figure supplement 1*.

**Figure supplement 1—source data 2.** Original files for gels are displayed in *Figure 3—figure supplement 1*.

showed that only PcrV (from *Pseudomonas aeruginosa*) was able to activate the NLRC4 inflammasome (*Figure 5—figure supplement 3B*). When the terminal five residues (T5R) of PcrV were deleted, the resulting ΔT5R mutant lost the capacity to activate NLRC4 (*Figure 5—figure supplement 3C*). LcrV, which did not activate NLRC4 (*Figure 5—figure supplement 3B*), shares a relatively high level (36.3%) of identity with PcrV but differs from PcrV in the T6R (*Figure 5—figure supplement 3D*). Substitution of the T6R of LcrV with the T5R of PcrV enabled the mutant, LcrV-T6RM, to gain the ability to activate NLRC4 inflammasome (*Figure 5—figure supplement 3D*). Similarly, the substitution of the T5R of EspA$_{EHEC}$ with the T6R of EseB enabled the mutant, EspA$_{EHEC}$-T5RM, to activate NLRC4 inflammasome (*Figure 5—figure supplement 3E*).

## Discussion

In this study, we examined the mechanism of inflammasome-mediated pyroptosis induced by bacterial T3SS translocon. Well-known inflammasome proteins, such as NLRP3 and NLRC4, and Casp4 are intracellular PRRs that induce pyroptosis during intracellular bacterial infections (*Storek and Monack, 2015*; *Broz and Dixit, 2016*). While NLRP3 activation can occur upon alterations in cellular homeostasis (*Gong et al., 2018*; *Hughes and O'Neill, 2018*; *Seoane et al., 2020*), NLRC4 and Casp4 are primarily activated by specific PAMP ligands presented by microbial organisms (*Christgen et al., 2020*; *Egan et al., 2023*). Previous studies have demonstrated that *E. tarda* components or secreted particles can cause pyroptosis in murine macrophages and human epithelial cells (*Xie et al., 2014*; *Zhang et al., 2016*; *Chen et al., 2017*; *Chen et al., 2018*; *Xu et al., 2018*). In this study, we found that in the infection model of THP-1 derived human macrophages, which express multiple inflammasomes, *E. tarda* induced pyroptosis in a manner that depended on NLRC4, NLRP3, ASC, Casp1, and Casp4. This observation indicated that *E. tarda* infection activated both the canonical and the non-canonical inflammasome pathways, which might be due to the multiplicity of virulence factors expressed by *E. tarda*. In line with this result, cell death was nearly blocked when the bacteria were prevented from entering the host cell cytoplasm, suggesting that *E. tarda*-triggered cell death was an event that required direct interaction between the bacteria and the inflammasome molecules.

T3SS is one of the critical armaments of intracellular bacteria to combat the host's defense system (*Portaliou et al., 2016*; *Deng et al., 2017*). T3SS delivers multiple bacterial effectors into host cells, which manipulate host immune responses to foster bacterial survival and expansion (*Raymond et al., 2013*; *Ratner et al., 2017*). This feature makes the T3SS apparatus readily exposed to the host cytosol, thus susceptible to detection by host receptors such as inflammasomes (*Miao et al., 2010*; *Zhao et al., 2011*; *Reyes Ruiz et al., 2017*; *Kofoed and Vance, 2011*; *Ratner et al., 2017*). Studies have demonstrated that in mice, T3SS needle protein, T3SS rod protein, and flagellin are directly detected by NAIP1, NAIP2, and NAIP5/6, respectively (*Zhao et al., 2011*; *Kofoed and Vance, 2011*). In contrast, in humans, T3SS needle/rod proteins and flagellin are all detected by a single NAIP (*Reyes Ruiz et al., 2017*; *Yang et al., 2013*; *Kortmann et al., 2015*). For *E. tarda*, two reports showed that flagellin was required to induce fish macrophage death (*Xie et al., 2014*) but not required to induce murine macrophage death (*Chen et al., 2017*). In our study, we found that flagellin was dispensable for *E. tarda*-induced pyroptosis of human macrophages. Consistently, the flagellin proteins were unable to activate human NAIP/NLRC4 inflammasomes. This observation differs from that of other pathogen flagellin proteins, which are recognized by NLRC4 inflammasomes (*Zhao et al., 2011*; *Reyes Ruiz et al., 2017*; *Kortmann et al., 2015*). Like flagellin, the *E. tarda* rod protein Esal also failed to activate the NLRC4 inflammasome. These results indicate that *E. tarda* flagellin and rod evade recognition by human NLRC4, probably as a strategy to facilitate bacterial infection. Unlike the needle, rod proteins, and flagellin, T3SS translocon proteins have seldom been reported to promote inflammasome activation and pyroptosis. Limited studies showed that the *Yersinia* translocon proteins YopD and YopB could translocate into host cells and indirectly activate inflammasome, resulting in cell death (*Casson et al., 2013*; *Zwack et al., 2015*; *Zwack et al., 2017*); the translocon proteins of *Pseudomonas*

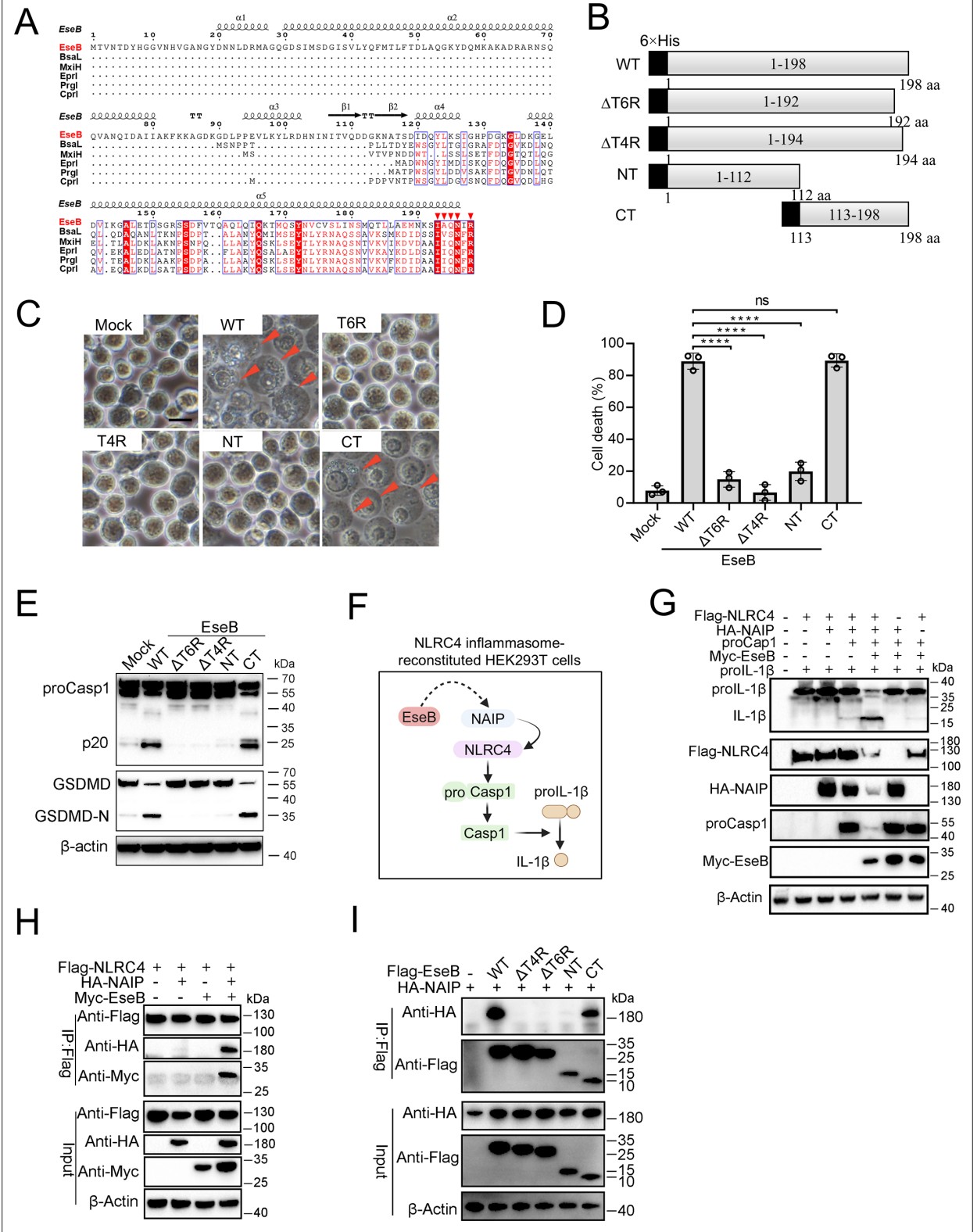

**Figure 4.** Identification of the functional important region in EseB. (**A**) Sequence alignment of EseB and T3SS needle proteins with NLRC4/NAIP-stimulating activity. (**B**) A diagram showing EseB wild-type (WT) and truncates. (**C–E**) THP-1 cells were electoporated with or without (Mock) EseB WT or truncated. The cells were subjected to microscopy (**C**), cell death analysis (**D**), and immunoblot with antibodies against Casp1, GSDMD, and β-actin (loading control) (**E**). In (**C**), red arrowheads indicate pyroptotic cells; scale bar, 10 μm. For panel D, data are the means of triplicate assays and are shown

*Figure 4 continued on next page*

*Figure 4 continued*

as means ± SD. ns, not significant, ****p<0.0001, one-way ANOVA with Dunnett's multiple-comparison test. (**F**) A diagram showing the detection of the activating effect of EseB on NAIP/NLRC4 in NLRC4 inflammasome-reconstituted HEK293T cells by determining proIL-1β cleavage. (**G**) HEK293T cells were transfected with or without the indicated combination of vectors expressing Flag-tagged NLRC4, HA-tagged NLR-family apoptosis inhibitory protein (NAIP), Myc-tagged EseB, proCasp1, and proIL-1β for 24 hr. The cells were subjected to immunoblot using antibodies against the tags or the proteins with β-actin as a loading control. (**H**) HEK293T cells were transfected with the indicated combination of vectors expressing Flag-tagged NLRC4, HA-tagged NAIP, and Myc-tagged EseB. The cells were subjected to immunoprecipitation (IP) using antibodies against the tags with β-actin as a loading control. (**I**) HEK293T cells were transfected with the indicated combination of vectors expressing HA-tagged NAIP and Flag-tagged EseB variants. IP was performed as above.

The online version of this article includes the following source data and figure supplement(s) for figure 4:

**Source data 1.** PDF file containing the original blots for *Figure 4E*.

**Source data 2.** Original files for blots are displayed in *Figure 4E*.

**Source data 3.** PDF file containing the original blots for *Figure 4G*.

**Source data 4.** Original files for blots are displayed in *Figure 4G*.

**Source data 5.** PDF file containing the original blots for *Figure 4H*.

**Source data 6.** Original files for blots are displayed in *Figure 4H*.

**Source data 7.** PDF file containing the original blots for *Figure 4I*.

**Source data 8.** Original files for blots are displayed in *Figure 4I*.

**Source data 9.** The numerical source data corresponds to *Figure 4*.

**Figure supplement 1.** SDS-PAGE analysis of purified recombinant proteins.

**Figure supplement 1—source data 1.** PDF file containing the original gel for *Figure 4—figure supplement 1*.

**Figure supplement 1—source data 2.** Original file for gel is displayed in *Figure 4—figure supplement 1*.

**Figure supplement 2.** The ability of *Edwardsiella tarda* EseB, rod (EsaI), needle (EsaG), and flagellin (FliC1/2) to activate NLRC4.

**Figure supplement 2—source data 1.** PDF file containing the original blots for *Figure 4—figure supplement 2A*.

**Figure supplement 2—source data 2.** Original files for blots are displayed in *Figure 4—figure supplement 2A*.

**Figure supplement 2—source data 3.** PDF file containing the original blots for *Figure 4—figure supplement 2B*.

**Figure supplement 2—source data 4.** Original files for blots are displayed in *Figure 4—figure supplement 2B*.

**Figure supplement 2—source data 5.** PDF file containing the original blots for *Figure 4—figure supplement 2C*.

**Figure supplement 2—source data 6.** Original files for blots are displayed in *Figure 4—figure supplement 2C*.

*aeruginosa* PopB–PopD may be associated with inflammasome activation (*Dortet et al., 2018*). Currently, it is unknown whether any inflammasome can directly recognize the translocon proteins. In the present study, mutational analyses showed that the translocon proteins were essential for *E. tarda* to induce pyroptosis and activate the non-canonical inflammasome Casp4. In particular, we found that the translocon protein EseB, when present intracellularly, sufficed to cause pyroptosis via NLRC4/NAIP. Moreover, the terminal residues proved to be vital for EseB activity, and the CT region alone could trigger pyroptosis in a manner similar to EseB. Consistently, although EseB differs dramatically from the NLRC4/NAIP-stimulating needle proteins in the NT region, it shares notable identities with the needle proteins in the CT region, suggesting a conserved mechanism of inflammasome activation via the CT region in these proteins.

Although T3SS is present in many bacterial pathogens, the particular mechanisms of T3SS translocon proteins, notably EseB, to induce host immune response are unclear. In this study, we identified EseB homologs in a large number of bacteria with T3SS and found that, like *E. tarda* EseB, all of these identified EseB homologs were able to activate NLRC4/NAIP, thus suggesting the wide existence of bacteria–host interaction mediated by EseB and the NLRC4/NAIP inflammasome. Sequence analysis revealed highly conserved T6R in all of the EseB homologs, which was crucial to NLRC4 inflammasome activation. Similar observation has been reported for the needle protein of T3SS. It has been shown that deletion of the last five amino acids from the needle protein prevented its self-association, so

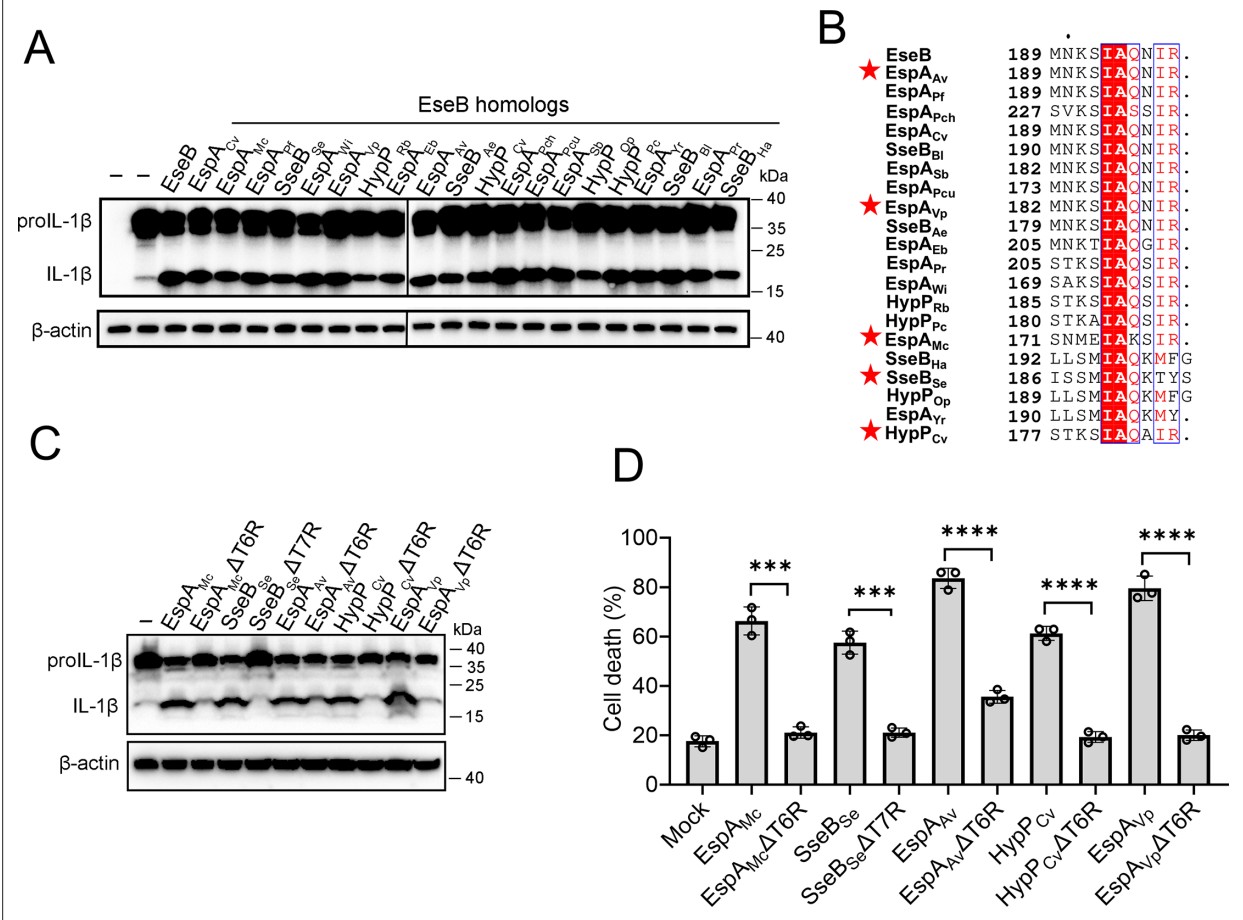

**Figure 5.** The ability of the EseB homologs to activate the NLRC4/NLR-family apoptosis inhibitory protein (NAIP) inflammasome. (**A**) NLRC4 inflammasome-reconstituted HEK293T cells were transfected with or without EseB homologs (*Supplementary file 2*). The cells were immunoblotted with antibodies against IL-1β and β-actin (loading control). (**B**) Sequence alignment of the C-terminal regions of the EseB homologs. Red stars indicate the EseB selected for mutation analysis. (**C**) NLRC4 inflammasome-reconstituted HEK293T cells expressing or not expressing the indicated EseB homologs or their mutants were immunoblotted as panel A. (**D**) THP-1 cells were electroporated with the indicated EseB homologs or their mutants or PBS (mock) and then determined for cell death. Data are the means of triplicate assays and are shown as means ± SD. ns, not significant, ***p<0.001, ****p<0.0001, Student's *t*-test.

The online version of this article includes the following source data and figure supplement(s) for figure 5:

**Source data 1.** PDF file containing the original blots for *Figure 5A*.

**Source data 2.** Original files for blots displayed in *Figure 5A*.

**Source data 3.** PDF file containing the original blots for *Figure 5C*.

**Source data 4.** Original files for blots are displayed in *Figure 5C*.

**Source data 5.** The numerical source data corresponds to *Figure 5*.

**Figure supplement 1.** Phylogenetic analysis of EseB homologs.

**Figure supplement 2.** SDS-PAGE analysis of purified recombinant proteins.

**Figure supplement 2—source data 1.** PDF file containing the original gels for *Figure 5—figure supplement 2*.

**Figure supplement 2—source data 2.** Original files for gels are displayed in *Figure 5—figure supplement 2*.

**Figure supplement 3.** The effects of translocator proteins on NLRC4/NLR-family apoptosis inhibitory protein (NAIP) inflammasome activation.

**Figure supplement 3—source data 1.** PDF file containing the original blots for *Figure 5—figure supplement 3B*.

**Figure supplement 3—source data 2.** Original files for blots are displayed in *Figure 5—figure supplement 3B*.

**Figure supplement 3—source data 3.** PDF file containing the original blots for *Figure 5—figure supplement 3C*.

**Figure supplement 3—source data 4.** Original files for blots are displayed in *Figure 5—figure supplement 3C*.

**Figure supplement 3—source data 5.** PDF file containing the original blots for *Figure 5—figure supplement 3D*.

*Figure 5 continued on next page*

*Figure 5 continued*

**Figure supplement 3—source data 6.** Original files for blots are displayed in *Figure 5—figure supplement 3D*.

**Figure supplement 3—source data 7.** PDF file containing the original blots for *Figure 5—figure supplement 3E*.

**Figure supplement 3—source data 8.** Original files for blots are displayed in *Figure 5—figure supplement 3E*.

that the protein could exist only in the monomeric form, indicating the importance of these terminal amino acids in maintaining the stability of the high-order structure of the protein (*Deane et al., 2006*; *Zhang et al., 2006*; *Wang et al., 2007*). In support of this, the cryo-electron microscopic structure of the needle–HsNAIP–HsNLRC4 complex showed that the last few amino acids of the needle were involved in interaction with human NAIP (*Matico et al., 2024*). The importance of the last few residues is also suggested by the report that *S. typhimurium* SPI2 T3SS rod protein SsaI may evade NLRC4/NAIP inflammasome recognition by alteration in the last eight amino acids (*Miao et al., 2010*). In our study, we found that the replacement of the terminal residues of EspA$_{EHEC}$ with that of EseB switched EspA$_{EHEC}$ to an NLRC4 activator. Similar observations were made with LcrV. These results suggest that the lack of EseB T6R-like terminal residues might be a strategy for EspA$_{EHEC}$ and LcrV to evade host immune detection. However, we also observed that IpaC and SipC, which possess EseB T6R-like terminal residues, failed to activate the NLRC4 inflammasome. Together these observations indicate that the terminal amino acids are a key, but not the sole, determinant in NLRC4 activation. Future studies are needed to find out the additional determinant(s) that work with the terminal residues to activate the NLRC4-mediated signaling.

In summary, our study demonstrated the importance and the mechanism of bacterial T3SS translocon in host interaction. We found that the translocon protein, EseB, triggered pyroptosis by directly activating the host NLRC4/NAIP inflammasome via the CT region, especially the T6R. Both the sequence and the inflammasome-stimulating function of the T6R were highly conserved in the EseB homologs of diverse bacteria. However, it must be said that for a translocator protein, the possession of a conserved T6R alone does not suffice to activate the NLRC4 inflammasome. These findings deepened the understanding of the function and mechanism of T3SS-mediated interaction between pathogens and hosts.

## Materials and methods

### Key resources table

| Reagent type (species) or resource | Designation | Source or reference | Identifiers | Additional information |
|---|---|---|---|---|
| Gene (*Edwardsiella tarda* and others) | Translocon genes | GenBank, UniProtKB | | *Supplementary files 1 and 2* |
| Strain, strain background (*E. tarda*) | E. tarda | *Li et al., 2022* | | |
| Strain, strain background (*E. tarda*) | ΔeseB | This paper | | In-frame deletion of *eseB* |
| Strain, strain background (*E. tarda*) | ΔescA | This paper | | In-frame deletion of *escA* |
| Strain, strain background (*E. tarda*) | ΔeseC | This paper | | In-frame deletion of *eseC* |
| Strain, strain background (*E. tarda*) | ΔeseD | This paper | | In-frame deletion of *eseD* |
| Strain, strain background (*E. tarda*) | ΔescB-D | This paper | | In-frame deletion of *eseB-eseD* |
| Strain, strain background (*E. tarda*) | ΔfliC | This paper | | In-frame deletion of *fliC1/2* |
| Strain, strain background (*E. coli*) | BL21(DE3) | TransGen Biotech | CD601 | |
| Cell line (*Homo sapiens*) | HEK293T | ATCC | Cat# CRL-3216, RRID:CVCL_0063 | |

*Continued on next page*

*Continued*

| Reagent type (species) or resource | Designation | Source or reference | Identifiers | Additional information |
|---|---|---|---|---|
| Cell line (*H. sapiens*) | THP-1 | Cell Resource Center, IBMS, CAMS/PUMC | 1101HUM-PUMC000057, RRID:CVCL_0006 | |
| Cell line (*H. sapiens*) | THP1-Null | InvivoGen | thp-null | Control cells |
| Cell line (*H. sapiens*) | THP1-Casp1-KD | InvivoGen | thp-dcasp1 | *Casp1* knockdown |
| Cell line (*H. sapiens*) | THP1-NLRP3-KD | InvivoGen | thp-dnlp | *Nlrp3* knockdown |
| Cell line (*H. sapiens*) | THP1-NLRC4-KO | This paper | | *Nlrc4* knockout |
| Cell line (*H. sapiens*) | THP1-Casp4-KO | This paper | | *Casp4* knockout |
| Cell line (*H. sapiens*) | THP1-Aim2-KO | This paper | | *Aim2* knockout |
| Cell line (*H. sapiens*) | THP1-ASC -KO | **Wang et al., 2024** | | *ASC* knockout |
| Cell line (*H. sapiens*) | THP1-GSDMD-KO | **Zhao et al., 2021** | | *Gsdmd* knockout |
| Transfected construct (*H. sapiens*) | shRNA-NAIP | This paper | | Lentiviral construct for *NAIP* knockdown |
| Transfected construct (*H. sapiens*) | sgRNA-Aim2, Casp4, NLRC4 | This paper | | Lentiviral construct for *Aim2, Casp4, NLRC4 gene* knockout |
| Antibody | anti-Caspase-1 (Rabbit polyclonal) | Cell Signaling Technology | Cat# 2225, RRID:AB_2243894 | WB (1:1000) |
| Antibody | anti-GSDMD (Rabbit polyclonal) | Cell Signaling Technology | Cat# 96458, RRID:AB_2894914 | WB (1:1000) |
| Antibody | anti- IL-1β (Rabbit monoclonal) | Cell Signaling Technology | Cat# 12703, RRID:AB_2737350 | WB (1:1000) |
| Antibody | anti-Caspase-4 (Rabbit monoclonal) | Cell Signaling Technology | 42264T | WB (1:1000) |
| Antibody | anti-6×His tag mAb (Rabbit monoclonal) | Abcam | ab213204 | WB (1:1000) |
| Antibody | anti-flag-tag (Rabbit monoclonal) | ABclonal | Cat# AE063, RRID:AB_2771920 | WB (1:1000) |
| Antibody | anti-HA-Tag (Mouse monoclonal) | ABclonal | Cat# AE008, RRID:AB_2770404 | WB (1:1000) |
| Antibody | anti-Myc-Tag (Mouse monoclonal) | ABclonal | Cat# AE010, RRID:AB_2770408 | WB (1:1000) |
| Antibody | anti-β-actin (Mouse monoclonal) | ABclonal | Cat# AC004, RRID:AB_2737399 | WB (1:1000) |
| Antibody | HRP goat anti-mouse IgG | ABclonal | Cat# AS003, RRID:AB_2769851 | WB (1:1000) |
| Antibody | HRP goat anti-rabbit IgG | Abcam | Cat# ab97051, RRID:AB_10679369 | WB (1:1000) |
| Recombinant DNA reagent | pLKO.1 puro | Addgene | 8453, RRID:Addgene_8453 | Negative control lentiviral construct |
| Recombinant DNA reagent | pDM4 | This paper | | The suicide plasmid |
| Recombinant DNA reagent | pET-28a | Novagen | 69864 | |
| Recombinant DNA reagent | pCS2Flag | Addgene | 16331, RRID:Addgene_16331 | |
| Sequence-based reagent | PCR primers | This paper | | *Supplementary file 1* |
| Commercial assay or kit | CytoTox 96 Non-Radioactive Cytotoxicity Assay kit | Promega | G1780 | |
| Commercial assay or kit | Human IL-1β ELISA kit | NeoBioscience | EHC002B | |
| Chemical compound, drug | cytochalasin B | Abcam | ab143482 | |
| Chemical compound, drug | cytochalasin D | Invitrogen | PHZ1063 | |
| Software, algorithm | Prism 10 | GraphPad | RRID:SCR_002798 | https://www.graphpad.com/ |

## Cells and cell culture

HEK293T (CRL-3216) and THP-1 (1101HUM-PUMC000057) cells were purchased from American type culture collection, ATCC and Cell Resource Center, IBMS, CAMS/PUMC, respectively. The cells

were maintained at 37 °C in a 5% $CO_2$ humidified incubator. HEK293T cells were cultured in DMEM (C11995500, Gibco) supplemented with 10% (v/v) FBS (10099–141 C, Gibco), 1% penicillin, and streptomycin (SV30010, HyClone). THP-1 cells were cultured in complete RPMI 1640 medium composed of RPMI 1640 (C22400500, Gibco) medium supplemented with 10% (v/v) FBS and 1% penicillin and streptomycin. THP1-Null (control) (thp-null), THP1-Casp1-KD (*Casp1* knockdown) (thp-dcasp1), and THP1-NLRP3-KD (*Nlrp3* knockdown) (thp-dnlp) were obtained from InvivoGen and maintained as instructed by the manufacturer. All cell lines were authenticated by STR profiling and confirmed to be free of mycoplasma contamination.

## Gene knockout and knockdown

THP-1 cells with gene knockout were generated using the CRISPR-Cas9 system as described previously (*Zhao et al., 2021*; *Wang et al., 2024*). Briefly, the sgRNAs targeting Aim2 (5'- TTCACGTT TGAGACCCAAGA-3'), Casp4 (5'-TGGTGTTTTGGATAACTTGG-3'), and NLRC4 (5'-CCACTACCACTG AGTGCCTG-3') were used for lentiviral constructs. The cells were treated with the lentiviral constructs and selected with puromycin. After selection, the gene knockout cells derived from single cells were further verified by PCR and sequence analysis. For *NAIP* gene knockdown via short hairpin RNA (shRNA), the oligo targeting *NAIP* (5'-CCGGGCCGTGGTGAACTTTGTGAATCTCGAGATTCACAAA GTTCACCACGGCTTTTTG-3' and 5'-AATTCAAAAAGCCGTGGTGAACTTTGTGAATCTCGAGATT CACAAAGTTCACCACGGC-3') was cloned into pLKO.1 puro (8453, Addgene), which was then used for lentiviral construct as above. The pLKO.1 scramble (1864, Addgene) was used for creating the negative control lentiviral construct. THP-1 cells were treated with the lentiviral constructs and selected as above. The knockdown efficiency was verified by RT-PCR with primers F (5'-GGCCAAAC TGATCATCCAGC-3') and R (5'-TGGCATGTTGTCCAGTGCTT-3').

## Bacterial strains and culturing

The *E. tarda* mutants with markerless in-frame deletion of *eseB*, *escA*, *eseC*, *eseD*, *eseB-eseD*, and *fliC1/2* (Δ*eseB*, Δ*escA*, Δ*eseC*, Δ*eseD*, Δ*eseB-D* and Δ*fliC*, respectively) were constructed as reported previously (*Li et al., 2022*; *Liu et al., 2022*). Briefly, the fragments upstream and downstream of the target gene were amplified by overlapping PCR and inserted into the suicide plasmid pDM4. The recombinant plasmids were introduced into *E. tarda*, and mutant strains were generated by a two-step homologous recombination. The deletion of the target gene was confirmed by PCR and sequence analysis of the PCR products. The information on primers and the mutants is shown in *Supplementary file 1*. *E. tarda* and its mutants were grown in Luria–Bertani (LB) medium supplemented with 20 µg/mL polymycin B (P8350, Solarbio) at 28 °C.

## Purification of recombinant proteins

The coding sequences of EscA, EseB, EseC, and EseD were amplified by PCR from the genome of *E. tarda*. All PCR products were cloned into the plasmid pET-28a (Novagen, 69864). *E. coli* BL21(DE3) (CD601, TransGen Biotech) was transformed with each of the recombinant plasmids, and the transformant was grown in LB medium at 37 °C until $OD_{600}$ 0.6. Isopropyl-β-D-thiogalactopyranoside (I8070, Solarbio) (0.2 mM) was added to the bacterial culture, and the culture was continued overnight at 16 °C. Bacteria were collected and lysed in Buffer A (20 mM Tris-HCl pH 8.0, 300 mM NaCl, and 10 mM imidazole). The recombinant proteins with His-tag were purified with Ni-NTA Agarose (30210, Qiagen). The proteins loaded onto the Ni-NTA column were washed with 60 column volumes of the Buffer B (20 mM Tris-HCl pH 8.0, 300 mM NaCl, 40 mM imidazole, and 0.1% Triton X114), and then with 80 column volumes of Buffer C (20 mM Tris-HCl pH 8.0, 300 mM NaCl, and 40 mM imidazole). The proteins were finally eluted with Buffer D (20 mM Tris-HCl pH 8.0, 300 mM NaCl, and 250 mM imidazole) and dialyzed against Buffer E (20 mM Tris-HCl pH 8.0, and 150 mM NaCl). The purified proteins were subjected to SDS–PAGE. Protein concentrations were determined with the BCA Protein Assay Kit (P0010, Beyotime) according to the manufacturer's instructions.

## *E. tarda* infection in THP-1 cells

THP-1 cells were differentiated into macrophages with PMA overnight (*Zhao and Sun, 2022*). *E. tarda* variants were cultured in LB medium with 20 µg/mL polymycin B at 28 °C until $OD_{600}$ 0.8. The bacteria were washed with PBS twice and then mixed with the differentiated THP-1 cells at MOI = 10. The

mixture was centrifuged at 800 g for 8 min and incubated at 30 °C for 1 hr in a 5% $CO_2$ humidified incubator. To kill the extracellular bacteria, gentamycin (500 µg/mL) was added to the cells, followed by incubation for 0.5 hr. The culture medium was replaced with a fresh medium containing 40 µg/mL gentamycin. To prevent bacterial entry into cells, the cells were treated with 50 µM cytochalasin B (ab143482, Abcam) or 10 µM cytochalasin D (PHZ1063, Invitrogen) for 0.5 hr prior to infection.

## Protein electroporation

THP-1 cells were cultured in complete RPMI 1640 medium to a density of ~1×10⁶ cells/mL. The cells were washed with precooled cytoporation medium T (47–0002, BTXpress) twice and resuspended in medium T to 5×10⁶ cells/mL. Protein electroporation was performed using a Gemini X2 electroporator (45–2006, BTX) as follows. The protein (2 µg) was added to 1×10⁶ cells in 200 µl medium T. Electroporation was then performed at the setting of 300 V, 10 ms, and 1 pulse. The cells were transferred into 1 mL pre-warmed OPTI-MEM medium (31985070, Gibco) and incubated for 1 hr. Then both cell lysates and supernatants were collected as described previously (*Zhao et al., 2021*; *Zhao and Sun, 2022*) for immunoblotting. Cell death was determined with CytoTox 96 Non-Radioactive Cytotoxicity Assay kit (G1780, Promega).

## NLRC4 inflammasome reconstitution in HEK293T cells

The coding sequences of human NLRC4, NAIP, Casp1, and proIL-1β were cloned from PMA-differential THP-1 cells and inserted into pCS2Flag (16331, Addgene)-based expression vectors with different tags. The coding sequences of EseB homologs from various bacteria (GenBank accession numbers shown in *Supplementary file 2*) were synthesized by Sangong Biotech (Shanghai, China). For NLRC4 inflammasome reconstitution, HEK293T cells were seeded into six-well plates overnight and then transfected with the indicated combination of plasmids (2 µg, 100 ng, 100 ng, 25 ng, and 100 ng for the plasmids expressing proIL-1β, NLRC4, NAIP, Casp1, and EseB, respectively) using Lipofectamine 3000 (L3000015, Invitrogen). At 24 hr post-transfection, the cells were lysed using RIPA buffer containing protease inhibitors. The cell lysates were analyzed by immunoblotting as described below.

## Immunoblotting and immunoprecipitation

Immunoblot was performed as reported previously (*Zhao and Sun, 2022*) with the following antibodies: Caspase-1 antibody (2225 S, Cell Signaling Technology), GSDMD antibody (96458 S, Cell Signaling Technology), IL-1β rabbit mAb (12703 S, Cell Signaling Technology), Caspase-4 mAb (42,264T, Cell Signaling Technology), anti-6x His tag mAb (ab213204, Abcam), flag-tag rabbit mAb (AE063, ABclonal), mouse anti-HA-Tag mAb (AE008, ABclonal), mouse anti-Myc-tag mAb (AE010, ABclonal), β-actin mouse mAb (AC004, ABclonal), HRP goat anti-mouse IgG (H+L) (AS003, ABclonal), goat anti-rabbit IgG H&L (HRP) (ab97051, Abcam). For immunoprecipitation, HEK293T cells transfected with the indicated plasmids were lysed in IP lysis buffer (50 mM Tris-HCl pH 7.6, 150 mM NaCl, 1% triton X-100, and 1x protease inhibitor cocktail), followed by centrifugation at 14,000 g for 10 min to remove cell debris. The supernatants were mixed with equilibrated anti-FLAG M2 magnetic beads (M8823, sigma) according to the manufacturer's instructions.

## Data analysis and statistics

Data were analyzed primarily using the Prism 10 software (https://www.graphpad.com/). Statistical analysis was conducted using Student's *t*-test for comparing two sets of data and one-way ANOVA for comparing three or more sets of data. Significance was defined as *$p < 0.05$, **$p < 0.01$, ***$p < 0.001$ and ****$p < 0.0001$.

## Acknowledgements

This work was supported by the Science & Technology Innovation Project of Laoshan Laboratory (LSKJ202203000), the National Natural Science Foundation of China (31330081), and the National Key Research and Development Project of China (2018YFD0900500).

## Additional information

### Funding

| Funder | Grant reference number | Author |
|---|---|---|
| Science & Technology Innovation Project of Laoshan Laboratory | LSKJ202203000 | Li Sun |
| National Natural Science Foundation of China | 31330081 | Li Sun |
| National Key Research and Development Program of China | 2018YFD0900500 | Li Sun |

The funders had no role in study design, data collection and interpretation, or the decision to submit the work for publication.

### Author contributions

Yan Zhao, Conceptualization, Data curation, Formal analysis, Validation, Investigation, Methodology, Writing – original draft; Hanshuo Zhu, Data curation, Validation, Investigation, Methodology; Jinqian Li, Hang Xu, Formal analysis, Investigation; Li Sun, Conceptualization, Resources, Data curation, Formal analysis, Supervision, Funding acquisition, Project administration, Writing – review and editing

### Author ORCIDs

Yan Zhao ⓘ https://orcid.org/0009-0001-3874-7770
Hang Xu ⓘ https://orcid.org/0000-0002-1378-5922
Li Sun ⓘ https://orcid.org/0000-0001-7183-7148

Reviewer #1 (Public review): https://doi.org/10.7554/eLife.100820.3.sa1
Reviewer #2 (Public review): https://doi.org/10.7554/eLife.100820.3.sa2
Author response https://doi.org/10.7554/eLife.100820.3.sa3

## Additional files

### Supplementary files

Supplementary file 1. The primers used for the construction of *Edwardsiella tarda* translocon mutants.

Supplementary file 2. Translocator proteins used in this study.

MDAR checklist

### Data availability

All data generated or analyzed during this study are included in the manuscript, supporting files and source data files.

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
