## [Editor Report · eLife Assessment]

This **important** study shows that Type 3 secretion translocons in Edwardsiella tarda and other bacteria activate the NAIP-NLRC4 inflammasome. The data from cellular and biochemical experiments showing that EseB is required for activation of the NLRC4 inflammasome are **convincing**. This paper is broadly relevant to those investigating host-pathogen interactions in diverse organisms.

---

## [Referee Report · Reviewer #1 (Public review)]

Summary:

In this study, Zhao and colleagues investigate inflammasome activation by E. tarda infections. They show that E. tarda induces the activation of the NLRC4 inflammasome as well as the non-canonical pathway in human THP1 macrophages. Further dissecting NLRC4 activation, the find that T3SS translocon components eseB, eseC and eseD are necessary for NLRC4 activation, and that delivery of purified eseB is sufficient to trigger NAIP-dependnet NLRC4 activation. Sequence analysis reveals that eseB shares homology within the C-terminus with T3SS needle and rod proteins, leading the authors to test if this region is necessary for inflammasome activation. They show that the eseB CT is required and that it mediates interaction with NAIP. Finally, they that homologs of eseB in other bacteria also share the same sequence and that they can activate NLRC4 in a HEK293T cell overexpression system.

Strengths:

This is a very nice study that convincingly shows that eseB and its homologs can be recognized by the human NAIP/NLRC4 inflammasome. The experiments are well-designed, controlled and described, and the papers is convincing as a whole.

Weaknesses:

The authors need to discuss their study in the context of previous papers that have shown an important role for E. tarda flagellin in inflammasome activation and test whether flagellin and/or E. tarda T3SSs needle or rod can activate NLRC4.

The authors show that eseB and its homologs can activate NLRC4, but there are also other translocon proteins that are very different such as YopB or PopB. and share little homology with eseB. It would be nice to include a section comparing the different type 3 secretion systems. are there 2 different families of T3SSs, those that feature translocon components that are recognized by NAIP-NLRC4 and those that cannot be recognized?

Comments on revisions:

The authors have addressed my concern with additional experiments, which strengthen the authors' conclusions.

---

## [Referee Report · Reviewer #2 (Public review)]

Summary:

This work by Zhao et al. demonstrates the role of the Edwardsiella tarda type 3 secretion system translocon in activating human macrophage inflammation and pyroptosis. The authors show the requirement of both the bacterial translocon proteins and particular host inflammasome components for E. tarda-induced pyroptosis. In addition, the authors show that the C-terminal region of the translocon protein, EseB, is both necessary and sufficient to induce pyroptosis when present in the cytoplasm. The most terminal region of EseB was determined to be highly conserved among other T3SS-encoding pathogenic bacteria and a subset of these exhibited functionally similar effects on inflammasome activation. Overall, the data support the conclusions and interpretations and provide valuable insights into interactions between bacterial T3SS components and the host immune system., thereby expanding our understanding of E. tarda pathogenesis.

Strengths:

The authors use established and reliable molecular biology and bacterial genetics strategies to characterize the roles of the bacterial T3SS translocon and host inflammasome pathways to E. tarda-induced pyroptosis in human macrophages. These observations are naturally expanded upon by demonstrating the specific regions of EseB that are required for inflammasome activation and the conservation of this sequence and function among other pathogenic bacteria.

---

## [Author Response]

The following is the authors’ response to the original reviews.

**Reviewer #1 (Public Review):**
Summary:In this study, Zhao and colleagues investigate inflammasome activation by E. tarda infections. They show that E. tarda induces the activation of the NLRC4 inflammasome as well as the non-canonical pathway in human THP1 macrophages. Further dissecting NLRC4 activation, they find that T3SS translocon components eseB, eseC and eseD are necessary for NLRC4 activation and that delivery of purified eseB is sufficient to trigger NAIP-dependent NLRC4 activation. Sequence analysis reveals that eseB shares homology within the C-terminus with T3SS needle and rod proteins, leading the authors to test if this region is necessary for inflammasome activation. They show that the eseB CT is required and that it mediates interaction with NAIP. Finally, they that homologs of eseB in other bacteria also share the same sequence and that they can activate NLRC4 in a HEK293T cell overexpression system.Strengths:This is a very nice study that convincingly shows that eseB and its homologs can be recognized by the human NAIP/NLRC4 inflammasome. The experiments are well designed, controlled and described, and the papers is convincing as a whole.Weaknesses:The authors need to discuss their study in the context of previous papers that have shown an important role for E. tarda flagellin in inflammasome activation and test whether flagellin and/or E. tarda T3SSs needle or rod can activate NLRC4.The authors show that eseB and its homologs can activate NLRC4, but there are also other translocon proteins that are very different such as YopB or PopB. and share little homology with eseB. It would be nice to include a section comparing the different type 3 secretion systems. are there 2 different families of T3SSs, those that feature translocon components that are recognized by NAIP-NLRC4 and those that cannot be recognized?(1) The authors need to discuss their study in the context of previous papers that have shown an important role for E. tarda flagellin in inflammasome activation and test whether flagellin and/or E. tarda T3SSs needle or rod can activate NLRC4.

According to the reviewer’s suggestion, we added the relevant discussion (lines 326-334) and carried out additional experiments to examine whether *E. tarda* flagellin, needle, and rod could activate NLRC4. The relevant results are shown in Figure S3, Figure S5, and lines 226-230 and 269-274.

(2) The authors show that eseB and its homologs can activate NLRC4, but there are also other translocon proteins that are very different such as YopB or PopB. and share little homology with eseB. It would be nice to include a section comparing the different type 3 secretion systems. are there 2 different families of T3SSs, those that feature translocon components that are recognized by NAIP-NLRC4 and those that cannot be recognized?

According to the reviewer’s suggestion, additional experiments were performed to examine the NLRC4-activating potentials of 14 translocator proteins that share low sequence identities with EseB. The relevant results and discussion are shown in Figure S8 and lines 289-301; 364-372, and 377-379.

**Reviewer #2 (Public Review):**
Summary:This work by Zhao et al. demonstrates the role of the Edwardsiella tarda type 3 secretion system translocon in activating human macrophage inflammation and pyroptosis. The authors show the requirement of both the bacterial translocon proteins and particular host inflammasome components for E. tarda-induced pyroptosis. In addition, the authors show that the C-terminal region of the translocon protein, EseB, is both necessary and sufficient to induce pyroptosis when present in the cytoplasm. The most terminal region of EseB was determined to be highly conserved among other T3SS-encoding pathogenic bacteria and a subset of these exhibited functionally similar effects on inflammasome activation. Overall, the data support the conclusions and interpretations and provide interesting insights into interactions between bacterial T3SS components and the host immune system.Strengths:The authors use established and reliable molecular biology and bacterial genetics strategies to characterize the roles of the bacterial T3SS translocon and host inflammasome pathways to E. tarda-induced pyroptosis in human macrophages. These observations are naturally expanded upon by demonstrating the specific regions of EseB that are required for inflammasome activation and the conservation of this sequence among other pathogenic bacteria.Weaknesses:The functional assessment of EseB homologues is limited to inflammasome activation at the protein level but does not include the effects on cell viability as shown for E. tarda EseB. Confirmation that EseB homologues have similar effects on cell death would strengthen this portion of the manuscript.

According to the reviewer’s suggestion, the effects of representative EseB homologs on cell death were examined in the revised manuscripts (Figure 5D, Figure S7 and line 289).

**Recommendations for the authors:**

**Reviewer #1 (Recommendations For The Authors):**
I only have a few suggestions on how to improve the study:Activation of caspase-4 requires entry into the host cytosol. Can this be observed with E. tarda and is it T3SS dependent? The fact that deleting the translocon components abrogates all GSDMD activation (see Fig. 2D) suggests that also Casp4 activation requires an active T3SS. It would be useful for the reader to include some more information on the cellular biology of E. tarda.

In our study, we found that *E. tarda* could enter THP-1 cells (Figure S1), and host cell entry was not affected by deletion of *eseB-D* (Δ*eseB-D*) in the T3SS system (Figure 2B, C). Additional experiments showed that Δ*eseB-D* abolished the ability of *E. tarda* to activate Casp4 (Figure S2), implying that Casp4 activation required an active T3SS. Relevant changes in the revised manuscript: lines 223 and 224, 341-342.

The data presented by the authors suggest that escB is sensed by NLRC4 when overexpressed, they do however not prove that during an infection escB is the main factor that drives NLRC4 activation, since deficiency in escB also abrogated translocation of other potential activators of NLRC4, e.g. flagellin and T3SS needle and rod subunits. I would thus find it essential to properly test if E. tarda flagellin can activate NLRC4 by comparing a WT and flagellin deficient strain, and/or by transfecting or expressing E.t. flagellin in these cells, as well as testing whether E.t. rod and needle subunits act as NLRC4 activators. This is important as previous studies suggested that flagellin is the main activator of cytotoxicity during E. tarda infection.

Previous studies have shown that flagellin is required for *E. tarda*-induced macrophage death in fish [1] but not in mice [2]. In the revised manuscript, we performed additional experiments to examine whether *E. tarda* flagellin, needle, and rod could activate NLRC4. The relevant results are shown in Figure S3, Figure S5, and lines 226-230 and 269-274, and 326-334.

References

(1) Xie HX, Lu JF, Rolhion N, Holden DW, Nie P, Zhou Y, et al. *Edwardsiella tarda*-induced cytotoxicity depends on its type III secretion system and flagellin. Infect Immun. 2014;82(8):3436-45. doi: 10.1128/IAI.01065-13.

(2) Chen H, Yang D, Han F, Tan J, Zhang L, Xiao J, et al. The bacterial T6SS effector EvpP prevents NLRP3 inflammasome activation by inhibiting the Ca^2+^-dependent MAPK-JNK pathway. Cell Host Microbe. 2017;21(1):47-58. doi: 10.1016/j.chom.2016.12.004.

Figure 5/S4, please list the names of the eseB homologs. It is cumbersome to have to access GenBank with the accession number to be able to understand what proteins the authors define as homologs of eseB.

The names were added to the revised Table S2, Figure 5 and Figure S6 (the original Figure S4).

The authors mention that other translocon proteins, such as YopB/D and PopB/D, were suggested to cause inflammasome activation. How do these compare to eseB and its homologs? Do they share the CT motif?

Additional experiments were performed to compare the inflammasome activation abilities of EseB and other translocator proteins including YopD and PopD. The relevant results and discussion are shown in Figure S8 and lines 289-301, 364-372, and 377-379.

It would be nice to show that there are potentially two groups of translocon proteins, one group sharing homology to needle subunits within the CT region and another that is different. A quick look at the sequence of these proteins suggests that they are quite different and much larger than eseB.

In our study, additional experiments with more translocator proteins indicated that the possession of EseB T6R-like terminal residues does not necessarily guarantee the protein to activate the NLRC4 inflammasome. Relevant results and discussion are shown in lines 289-301, 364-372, and 377-379.